# The Physicochemical Attributes, Volatile Compounds, and Antioxidant Activities of Five Plum Cultivars in Sichuan

**DOI:** 10.3390/foods12203801

**Published:** 2023-10-17

**Authors:** Zixi Lin, Binbin Li, Maowen Liao, Jia Liu, Yan Zhou, Yumei Liang, Huaiyu Yuan, Ke Li, Huajia Li

**Affiliations:** 1Institute of Agriculture Products Processing Science and Technology, Sichuan Academy of Agriculture Science, Chengdu 610039, China; linzx1011@126.com (Z.L.); liaomaowen2022@163.com (M.L.); zhouyan199581@163.com (Y.Z.); liangym623@163.com (Y.L.); yuanhuaiyu53@163.com (H.Y.); 2Institute of Agricultural Products Processing Research, Xinjiang Academy of Agricultural Reclamation Sciences, Shihezi 832000, China; 15299660073@163.com; 3Horticulture Research Institute, Sichuan Academy of Agricultural Sciences, Chengdu 610039, China; nky_lj@163.com; 4College of Food Science, Sichuan Agricultural University, Ya’an 625014, China

**Keywords:** plum, physicochemical characteristics, antioxidant capacity, volatile metabolites, biomarkers

## Abstract

Plum (*Prunus salicina* Lindl.) is an important stone fruit crop in Sichuan that is increasingly in demand by consumers owing to its flavor and outstanding nutraceutical properties. The physicochemical characteristics, antioxidant capacity, and volatile profiles of five traditional and new plum cultivars in Sichuan were determined using high-performance liquid chromatography and gas chromatography time-of-flight mass spectrometry. The results showed that all plums exhibited an appropriate quality profile for fresh consumption; the new cultivar ‘ZH’ exhibited the highest soluble solids content, sugar–acid ratio, total phenolic content, total flavonoid content, and antioxidant capacity. High sugar–low acid properties were observed in five plum cultivars. Sucrose was the main sugar, while quinic acid and malic acid were the main organic acids. The plums were rich in volatile compounds and had specific volatile characteristics. A total of 737 volatiles were identified in the plum fruit, and orthogonal partial least-squares discriminant analysis was employed to screen 40 differential volatiles as markers for cultivar distinction. These findings offer comprehensive information on the physicochemical characteristics, antioxidant capacity, and volatile profiles of plums.

## 1. Introduction

Plums are popular stone fruits belonging to the subgenus *Prunus* (family Rosaceae). Through decades of domestication and the ongoing selection of wild plums, there are more than 40 species of plums distributed worldwide [1]. The Japanese plum (*Prunus salicina*) is considered one of the most important fruit crops commercially grown in China, Spain, and the USA [2]. Around the world, over 12,014,000 tons were harvested in 2021 as reported by the Food and Agriculture Organization of the United Nations. Currently, China has become one of the largest producers of plums owing to its geographic and climatic diversity, from where 6,626,300 plums were produced [3]. Sichuan has rich plum germplasm resources, a wide range of varieties, and excellent plum quality. For example, the traditional cultivars ‘ailisi’, ‘cuihongli’, ‘qingcuili’, ‘xiangtianli’, and ‘xiangli’ and the newly introduced cultivars ‘zihuang’ and ‘shengxuepo’ have been successfully cultivated on a large scale. There is limited information regarding the quality characteristics of these plums.

Fruit flavor and nutrition are important factors in fruit quality. As delicious table fruits, plums are characterized by a pleasant taste and aroma. The composition and content of sugars and organic acids are critical contributors affecting fruit taste, and these characteristics are strongly influenced by the cultivar and fruit maturity [4]. In addition, volatile components are important secondary metabolites that are essential for the overall flavor and commercial value of fruits; thus, in the past decades, some studies have analyzed the aroma components of plums [5,6]. Esters are the most important class of compounds both qualitatively and quantitatively. Other volatile constituents vary greatly among different plum cultivars, resulting in unique aroma characteristics for each cultivar [7]. Gas chromatography time-of-flight mass spectrometry (GC-TOF-MS) analysis has been extensively adopted to profile volatile metabolites to associate a specific odor with the corresponding fruit [8].

In addition to their juiciness and tastiness, mature plums contain various bioactive components. Since the 1990s, there has been a sharp rise in interest in research involving plums due to their high concentrations of phenolic compounds, particularly flavonoids and a subclass of anthocyanins [9]. Data from several studies have suggested many health benefits of plum extract, including improved bone health and anti-inflammatory and antioxidant effects [4,9]. Phenolic compounds are used as functional ingredients to enhance the antioxidant capacity of fruits. Studies have confirmed that plums have a higher total antioxidant capacity than apples, grapes, and bananas, owing to their high total phenolic content [10,11]. Li et al. [12] reported that polyphenol extracts of plums possess the remarkable ability to inhibit xanthine oxidase and scavenge free radicals. Additionally, the taste, color, and antioxidant qualities of fruits are greatly influenced by the phenolic composition and concentration found in Japanese plums [13].

The local market demand for safe and high-quality fruits is increasing because consumers are aware of their characteristic taste and nutritional value [14]. In this regard, the characterization of the nutritional profile and flavor compounds of traditional plum cultivars is a response to these emerging consumer trends. In the past 20 years, although Sichuan and other plum-producing regions have made some progress in plum breeding studies and the number of plum cultivars released has increased, the flavor and quality of plum fruit vary greatly depending on the location, climate, and variety [15]. This study aimed to investigate the physicochemical, antioxidant, and volatile compound characteristics of five traditional and new plum cultivars from Sichuan using high-performance liquid chromatography (HPLC) and GC-TOF-MS. This study provides useful information on the characteristic qualities and aromas of five plums from Sichuan and a theoretical basis for the breeding and selection of good cultivars.

## 2. Materials and Methods

### 2.1. Fruit Materials

Five plum cultivars were collected from Deyang (Sichuan, China): xiangli (XL), xiangtian (XT), ailisi (AL), shengxuepo (SX), and zihuang (ZH) (Figure 1). A total of 450 plum fruit were harvested from the upper, middle, and lower locations of 5–6 trees, and 90 fruit were collected from each sample. Ripe plums of similar size and with no visible outer damage were chosen according to their color, firmness, and aroma, as well as the judgment of the local growers. After harvest, all plum samples were carried to the experiment room and immediately washed. Different varieties of plums were randomly divided into three replicates. Thirty fruit per variety were separated for laboratory analyses of pomological characteristics. The remaining plums were weighed and frozen in liquid N_2_, then stored at −80 °C.

### 2.2. Reagents

1,1-diphenyl-2-picrylhydrazyl (DPPH), 2,2′-azinobis (3-ethylbenzothiazoline-6-sulfonic acid) diammonium salt (ABTS), and the ferric reducing antioxidant power (FRAP) assay were purchased from Sigma-Aldrich (Saint-Quentin-Fallavier, France). Standards for sucrose, glucose, fructose, sorbitol, and quinic, malic, citric, and succinic acids were obtained from Yuanye Biotechnology Co., Ltd. (Shanghai, China). All of the other chemicals used were of analytical grade.

### 2.3. Soluble Solids Content (SSC), Titratable Acidity (TA), pH, and Maturity Index (MI)

Fresh juice was produced using a juicer (SUPOR SJ30, Shanghai, China) for use in the determination of SSC and TA. SSC was analyzed using a sugar refractometer (Atago PR-101R, Tokyo, Japan) [16]. TA was measured using an automatic titrator (794 Basic Titrino, Metrohm, Switzerland) and expressed as % citric acid [17]. pH was determined with a PHS-3E meter (Shanghai Leici Co., Ltd., Shanghai, China) [18]. MI values were calculated as the SSC/TA ratio.

### 2.4. Sugars and Organic Acids

#### 2.4.1. Extraction of Sugars and Organic Acids

Sugars and organic acids were extracted according to the method described by Kodagoda et al. [19], and their contents were measured as described by Orazem et al. [20]. Briefly, the enucleated fresh plums were frozen in liquid nitrogen and ground to a fine powder, and 1.0 g of fruit powder was brought to a volume of 20 mL using distilled water. The samples were mixed for 3 min using a vortex instrument (IKA-Labortechnik, Staufen, Germany) and then extracted using ultrasound for 30 min at room temperature (24 °C) followed by centrifugation at 8000 rpm for 10 min at 4 °C and filtration through 0.45 μm cellulose filters. The sugar and organic acid content of the samples was measured using HPLC using an Agilent 1260 (Agilent Technologies, Santa Clara, CA, USA).

#### 2.4.2. HPLC System

Sugars (sucrose, glucose, fructose, and sorbitol) were detected using an RID-10A refractive index detector (Shimadzu, Kyoto, Japan) and RCM-Monosaccharide Ca^2+^ column (100 × 7.8 mm × 8 mum) at 80 °C with an injection volume of 10 μL. The chromatographic separation of sugars was performed using twice-distilled water as the mobile phase, a 0.3 mL/min flow rate, and a 40 min run time. Organic acids (malic, quinic, oxalic, and succinic acids) were detected using a 2487 UV detector (Waters Corporation, Milford, MA, USA) set at 210 nm after separation with an Aminex HPX-87H column (300 mm × 7.8 mm × 9 mm) kept at 60 °C. A dilute sulfuric acid solution (0.05 mol/L) was used as the mobile phase, with a 0.5 mL/min flow rate and a 35 min run time.

### 2.5. Total Phenolic Content (TPC), Total Flavonoid Content (TFC), and Total Anthocyanin Content (TAC)

The TPC was extracted by the following method as reported previously [21]. Briefly, the plum powder (5 g) was extracted using 25 mL of methanol aqueous (80% *v*/*v*) and ultrasound three times at 35 °C for 30 min each time and then centrifuged at 10,000 rpm for 8 min. After extraction, the TPC was determined using the Folin–Ciocalteu method [22]. First, 5 mL of Folin–Ciocalteu (10% *v*/*v*) reagent was mixed with 1 mL of polyphenol extract and left for 5 min and then neutralized with 4 mL sodium carbonate (7.5% *w*/*w*) and reacted at 25 °C for 90 min. The absorbance of the sample was measured at 760 nm, and the results were expressed as milligram equivalents of gallic acid per 100 g of fresh weight (mg GAE/100 g).

TFC was measured using a colorimetric assay [18]. In detail, 1 mL of the sample extract was mixed with 0.3 mL NaNO_2_ (5% *m/v*) for 5 min at room temperature (24 °C). Subsequently, 0.3 mL of AlCl_3_ (10% *v*/*v*) and 2 mL of NaOH (1 mol/L) were added. The absorbance of the mixture was measured at 510 nm and compared with that of the catechin standard. The results were expressed as milligrams of catechin equivalent (CE) per 100 g of fresh weight (mg CE/100 g).

TAC was determined using a Plant ANTH ELISA Kit (Mlbio, Shanghai, China) [23]. The plum samples (1 g) were ground in liquid nitrogen, diluted 10-fold with phosphate-buffered saline (0.05 mol/L, pH 7.8), and subjected to centrifugation at 8000 rpm for 15 min. The supernatant, standards, and horseradish peroxidase (HRP)-labeled antibodies were added to antibody-coated wells, followed by incubation and washing. Based on the principle that tetramethylbenzidine (TMB) is converted into a blue color by peroxidase catalysis and to a final yellow color by the action of sulfuric acid, the TMB substrate acts as a color-developing agent [24]. The depth of color was positively correlated with the anthocyanin concentration in the sample. The absorbance of each sample was measured at 450 nm.

### 2.6. Antioxidant Activity

The extract was prepared as described by Yu et al. [13]. Briefly, 9.0 mL of methanol (*v*/*v*, 80%) was added to the powder (1.0 g) at room temperature (24 °C), extracted using ultrasonication (300 W, 30 min), and centrifuged at 8000 rpm for 15 min. Finally, the supernatant was stored at −20 °C for later experimental use.

The antioxidant capacity of the samples was studied using ABTS and DPPH assays (for measuring the free radical scavenger activity) and the FRAP method (for assessing the reducing power) [21]. The absorbance was determined at 750, 520, and 595 nm using a microplate spectrophotometer (Tecan Infinite M1000 Pro; Tecan Trading, Mannedorf, Switzerland) for the ABTS, DPPH, and FRAP assays, respectively. Finally, all results were expressed in µmol Trolox equivalents per 100 g of dry mass (µmol TEs/100 g).

### 2.7. Volatile Compounds

Volatiles in the plums were isolated according to the solid-phase microextraction (SPME) method previously described by Özdemir et al. [25]. Briefly, 1 mL of plum juice was mixed with 5 mL of a saturated aqueous sodium chloride solution in a 20 mL SPME glass vial. The mixed solution was kept at 60 °C for 30 min and then extracted with divinylbenzene, carboxene, and polydimethylsiloxane at the same temperature for 30 min using an SPME needle. The fibers were then inserted into a GC injector. The desorbed volatiles were processed using the splitless mode at 250 °C for 5 min.

GC-TOF-MS analysis was performed using an Agilent 8890A GC equipped with a LECO Pegasus^®^ 4D GC × GC-TOF mass spectrometer (LECO Corporation, St. Joseph, MI, USA) and a DB-Heavy Wax column (30 m × 250 μm × 0.5 μm). The injection-port temperature was 260 °C and high-purity helium (>99.999%) was used as the carrier gas at a rate of 1.0 mL/min. The initial temperature of the GC was 40 °C for 3 min, then increased to 100 °C at a ramp rate of 4 °C/min for 2 min, 3 °C/min up to 180 °C for 2 min, followed by a ramp rate of 10 °C/min to 250 °C for 6 min. The ion source used for the mass spectrometer was 250 °C and the electron energy was 70 eV. Mass spectra were obtained at a scan rate of 2.84 scans/s in the *m*/*z* 35–550 range.

### 2.8. Statistical Analysis

All data are presented as the mean values ± standard deviation. Significant differences (*p* < 0.05) among the samples were determined using ANOVA with Tukey’s post hoc test using SPSS 26.0 (IBM Corporation, Armonk, NY, USA). Bar charts, heat maps, and radar charts were generated using GraphPad Prism 9.0 (GraphPad software, San Diego, CA, USA), and orthogonal partial least-squares discriminant analysis (OPLS-DA) was performed using SIMCA 14.1 (Umetrics AB, Umea, Sweden).

## 3. Results and Discussion

### 3.1. SSC, TA, pH, and MI of the Five Plum Cultivars

Researchers have found that the physicochemical properties of fruits are essential factors determining harvest maturity and consumer acceptance [19]. The values of SSC, TA, pH, and MI (SSC/TA) were calculated for the local cultivars ‘XL’, ‘AL’, and ‘XT’, as well as for the introduced cultivars ‘ZH’ and ‘SX’ (Figure 2), and statistically significant differences were found. The SSC values varied from 11.67 to 20.23 °Brix, with a mean of 15.53 °Brix, which was higher than the lowest value (8 °Brix) set by the European Union for certain drupes [26]. The pH ranged from 3.43 to 3.73 with a mean value of 3.56. These values were close to the pH values of peaches (3.37–4.62) and nectarines (3.05–4.47) [27]. Abidi et al. [26] suggested that ripened fruit with a pH higher than 4.0 was considered as non-acid. Therefore, all five plum cultivars showed acid content, as reported by Sifat et al. [28]. The five plum cultivars showed TA values ranging from 0.82% to 1.38%, with a mean value of 1.0%. Drogoudi and Pantelidis [29] found a wider TA range in European and Japanese plum cultivars (43 cultivars) (0.5–1.9%). The TA values of ‘ZH’ and ‘SX’ were greater than 1.0%, similar to an early-season plum (Black Amber) with a high TA [30]. In general, high TA values in fruits negatively affect consumer acceptance [31]. Interestingly, an “in-store” consumer test confirmed that, in the case of cultivars with SSC ≥ 12.0% resulting in high consumer acceptance, TA did not play a role at all. In the present study, only ‘AL’ had an SSC of less than 12 °Brix, and ‘XL’, ‘XT’, ‘SX’, and ‘ZH’ had the highest values (mean of 16.49 °Brix). The MI (SSC/TA) ranged from 12.66 to 19.43 among the five cultivars, with ‘ZH’ having the highest sugar–acid ratio. MI plays an important role in the organoleptic quality traits of mature fruits. The interaction between TA and SSC, rather than SSC alone, is a key factor in the consumer acceptance of plums. Crisosto et al. [30] found that plum cultivars with SSC < 12 °Brix and TA < 0.99% exhibited higher organoleptic attributes in alignment with consumer acceptance. Our results showed that only ‘AL’, with mean TA (0.82%) and SSC (11.67 °Brix) values, fell within this range.

### 3.2. Individual and Total Sugar Content in Five Plum Cultivars

The human sensory experience is driven by a combination of chemical constituents in a combined effect of primarily sugars, organic acids, and volatile compounds. The kinds and concentrations of organic acids and sugars have significant effects on the taste of ripe fruit [32]. Generally, fruit sweetness is positively correlated with consumer acceptance and is determined by total and individual sugar content [33]. However, there is little information on the pattern of sugar accumulation in the cultivars ‘ZH’, ‘SX’, ‘XL’, ‘XT’, and ‘AL’. There was diversity in the total and individual sugar content among the five plum cultivars (Figure 3a). The total sugar content ranged from 141.38 to 82.19 mg/g FW. Among them, ‘AL’ had the highest total sugar content, whereas ‘XL’ had the lowest. Except for the traditional cultivars ‘XL’ and ‘XT’, the total sugar contents of three plums were over 100 mg/g FW. In the fruit of each cultivar, four main soluble sugars, sucrose (28.15–58.37 mg/g FW), fructose (25.45–42.50 mg/g FW), glucose (18.30–57.11 mg/g FW), and sorbitol (0.72–1.80 mg/g FW), were identified and quantified, as previously reported [34]. Sucrose, fructose, glucose, and sorbitol accounted for 29.22–52.04%, 26.42–29.52%, 18.91–40.91%, and 0.94–1.31% of the total sugars, respectively (Figure 3b). These sugars have also been found in different fruits of plum cultivars in the genus *Prunus* [34,35]. Among the five tested plums, sucrose was the main source of sweetness. Similar patterns have been observed in flat peaches [36]. In contrast, the most abundant soluble sugars in cucumbers and sweet cherries are fructose and glucose [37,38]. In addition, the proportion of sugar components has a direct influence on the sweetness of the fruit. The ratios of sucrose to fructose to glucose were found to be approximately 5:2:1 in the cultivar ‘XT’ and 2:1:1 in ‘ZH’, ‘SX’, and ‘XL’, whereas a 1:1:2 ratio was observed in ‘AL’ in the current study. This may be the reason why different plum cultivars present a characteristic sweetness.

### 3.3. Individual and Total Organic Acid Content in the Five Plum Cultivars

The taste of plums is not influenced only by sugars, however. The diversity and concentration of organic acids are important contributors to their organoleptic properties [31]. In this study, the content of total organic acids was higher in the new cultivars ‘ZH’ and ‘SX’ (30.35 and 30.26 g/kg, respectively) compared to the traditional cultivars ‘XL’ and ‘XT’, which had the lowest (15.76 mg/g and 13.32 mg/g, respectively). Four major organic acids were found in the five plum cultivars: malic, quinic, succinic, and oxalic acids (Figure 3c). Previous studies have reported that malic and quinic acids account for the bulk of the organic acid content in ripe plums [35]. Similarly, quinic and malic acids were also the major acids contributing to the acidity of the five tested plums, with concentrations on an FW basis ranging from 6.86 mg/g (XT) to 12.55 mg/g (SX) and from 4.93 mg/g (XL) to 14.15 mg/g (SX), respectively; these accounted for 40.59% (AL) to 53.09% (XL) and 33.21% (ZH) to 46.78% (SX) of the total acid content, respectively (Figure 3d). These observations appear to be typical of the genus *Prunus* [35]. In addition, the content of succinic acid varied among the five cultivars and ranged from 11.03% to 18.61% of the total acid content. ‘ZH’ had three-fold higher succinic acid content than ‘XL’. Compared to other acids, oxalic acid was found in lower amounts in the five plum cultivars (approximately 0.94% of the total acids). The greatest content of oxalic acid was encountered in ‘ZH’ (0.43 mg/g) and the lowest content was in ‘XT’ (0.12 mg/g). Similar to the effect of sugar on sweetness, the acidity of plums is influenced by the proportions of individual organic acids. In detail, the ratio of malic to quinic acid was approximately 1:1 in all five plums, which may be the reason for the characteristic acidity of the taste of plums.

### 3.4. TPC, TFC, and TAC in the Five Plum Cultivars

The demand for high-quality fruits continues to increase, and plums have become popular because of their high nutritional value. Many studies indicate that plums have an effective preventive action against chronic and degenerative diseases, owing to their rich natural phenolic compounds, flavonoids, and anthocyanins [39]. However, there is limited information on the TPC, TFC, and TAC in these plum resources. TPC showed highly significant differences among the five plum cultivars (*p* < 0.05) (Figure 4a), and the values varied from 108.54 to 271.74 mg of GAE/100 g with an average of 197.00 mg of GAE/100 g. Similar trends but with higher values were reported previously for TPC in plum fruit (from cultivars such as Ruby Crunch, Ruby Red, PR03-34, PR04-32, and PR04-35, harvested from South Africa) extracted with a different solvent, methanol, with values ranging from 2.98 mg GAE/g to 3.83 mg GAE/g [15]. Kim et al. [39] reported that TPC ranged from 125.0 to 372.6 mg of GAE/100 g among eleven tested plum cultivars (harvested from Geneva). Gu et al. [40] proposed a variation in TPC owing to differences in fruit grown in different regions. The new cultivar ‘ZH’ had a higher TPC than the three traditional cultivars ‘XT’, ‘AL’, and ‘XL’. The new plum cultivars ‘ZH’ and ‘SX’ possessed more than twice the content of total phenols (271.74 and 217.58 mg of GAE/100 g, respectively) compared to the traditional cultivar ‘AL’ (108.54 mg of GAE/100 g). Ignat et al. [41] suggested that the high accumulation of TFC indicates more active production of phenolic compounds in mature fruits, which are more beneficial for resisting ultraviolet light and predators.

Polyphenols with flavonoid structures combine with aluminum chloride to produce a yellow solution that immediately turns red under alkaline conditions [41]. In this study, the range of TFC in different plums varied between 78.56 and 169.84 mg of CE/100 g (Figure 4b). The mean TFC among the five plum cultivars was 120.28 mg of CE/100 g. ‘ZH’ had the highest TFC, whereas ‘AL’ had the lowest. Compared to the results of Kim et al. [39], of a range (64.8–145.2 mg of CE/100 g) in eleven plum cultivars including Autumn Sweet, Beltsville Elite, Early Magic, Stanley, etc., Beltsville Elite B70197 had the highest TFC (257.5 mg of CE/100 g), demonstrating a significant difference (*p* < 0.05) among the selected fresh plums. Maietti et al. [42] explained many sources of variation in flavonoid content, such as fruit maturity, plant phenotypic status, and climatic conditions.

Anthocyanins are an important class of phenolic compounds that belong to the flavonoid class of plant secondary metabolites. Many studies have confirmed that anthocyanins are responsible for the color of many fruits and are economically important [43]. In this study, the TAC in five plum cultivars ranged from 7.63 to 35.24 pg/g, was determined by double-antibody sandwich assay (Figure 4c), and was inconsistent with the results of the pH time-difference method used by Gu et al. [44]. Currently, there is a lack of unified anthocyanin testing standards, and differences between experimental procedures cannot explain the inconsistencies in the existing literature [45]. In addition, anthocyanin accumulation is a key factor in determining the red fruit phenotype. According to Nour [46], anthocyanins are primarily found in the fruit peel and are responsible for the purple color of plums. Similar to our results, the new cultivar ‘ZH’ and the traditional cultivar ‘AL’ with purple peel showed significantly higher TAC levels than the other cultivars. The lowest TAC values were recorded for ‘XL’, ‘XT’, and ‘SX’, which were green- or yellow-skinned. In summary, ‘ZH’ is a new cultivar with high levels of TPC, TFC, and TAC that serves as an abundant source of phenolic compounds.

### 3.5. DPPH, ABTS, and FRAP in the Five Plum Cultivars

Antioxidants can form stable products by donating electrons to free radicals, thus preventing the chain reactions of free radicals and exerting antioxidant effects [21]. Taking into account the complexity of plum extracts and their mechanisms of action, three assays (DPPH, ABTS, and FRAP) were combined to evaluate plum antioxidant activity.

The DPPH assay is based on the hydrogen donor-mediated reduction of the stable radical DPPH to the yellow-colored diphenyl picrylhydrazine [47]. The findings suggested that the five plums tested affected DPPH radicals, and the differences were significant among the different cultivars (*p* < 0.05) (Figure 4d). The new cultivar ‘ZH’ showed the highest DPPH free radical scavenging activities with 27.16 µmol TEs/g. The three traditional cultivars ‘AL’, ‘XT’, and ‘XL’ showed lower DPPH values, but the differences were not significant among them (*p* > 0.05). Research suggests that the highest TPC corresponds to the strongest anti-free radical potential of the relevant fruit extracts, which is similar to our results [47,48].

Similar to DPPH, the ABTS assay is a widely available method for determining the anti-free radical scavenging ability of a sample based on its ability to transfer electrons or hydrogen atoms [21]. The ABTS^+^-reducing capability of the five plums differed significantly (*p* < 0.05) between 17.17 and 54.34 µmol TEs/g (Figure 4e). The new cultivar ‘ZH’ showed the highest ABTS-reducing capacity, followed by the ‘SX’ (43.41 µmol TEs/g) and ‘XL’ (34.33 µmol TEs/g) samples. In comparison, ‘XT’ and ‘AL’ exhibited relatively low ABTS radical scavenging abilities. Wu et al. [21] reported slightly lower ABTS radical scavenging activities in seven different peach varieties, ranging from 155.81 to 312.61 µmol TEs/100 g, as compared to our results. Similar to the study by Hameed et al. [47], the ABTS-reduction capacity of plums (0.47 ± 0.01 mg AAE/g) was higher than that of peaches (0.27 ± 0.02 mg AAE/g) and pears (0.12 ± 0.07 mg AAE/g). Tian and Schaich [49] proposed that potent ABTS free radical scavenging activity is closely related to the structure of phenols and flavonoids in the sample.

Unlike DPPH, the FRAP assay is based on the ability of a sample to donate electrons to reduce the ferric tripyridyltriazine complex to the ferrous tripyridyltriazine under low-pH conditions [50]. The FRAP value trends of the five plum extracts were similar to those of DPPH and ABTS, with the highest FRAP value of 32.19 µmol TEs/g recorded for the new cultivar ‘ZH’, while the two traditional cultivars ‘XT’ and ‘AL’ (7.75 and 10.77 µmol TEs/g, respectively) had the lowest values (Figure 4f). Kim et al. [39] determined the antioxidant activity of eleven plums using spectrophotometric methods, and the results ranged from 204.9 to 567.0 mg/100 g expressed as the equivalent antioxidant capacity of vitamin C. Lim et al. [51] also reported comparable results whereby Davidson’s plum exhibited the highest FRAP reducing power (500.38 ± 64.32 µmol Fe^2+^/g DW), followed by finger lime (46.16 ± 3.74 µmol Fe^2+^/g DW) and native pepperberry (48.48 ± 3.23 µmol Fe^2+^/g DW).

### 3.6. Volatile Profiles of the Five Plum Cultivars

#### 3.6.1. Analysis of Volatile Metabolites

Volatile organic compounds (VOCs) are key components of the fruit metabolome and provide a deeper understanding of the differences in aroma composition and their impacts on flavor perception [52]. An untargeted metabolic profiling approach was used to explore the volatile information using two-dimensional gas chromatography time-of-flight mass spectrometry (GC × GC-TOF-MS) for five selected plum cultivars. In total, 737 metabolites were identified. The volatile metabolites were analyzed and classified as 31 acids, 106 alcohols, 48 aldehydes, 75 alkanes, 93 alkenes, 111 aromatic compounds, 76 ketones, 108 esters, 9 furans, and 80 other compounds (Figure 5a). As shown in Figure 5b, alcohols, aromatics, and esters represented 15.25–19.09%, 15.77–19.38%, and 9.32–17.01% of the total VOCs, respectively, confirming that they were the major VOCs in the five plums. This is similar to the results of Cuevas et al. [52] who investigated VOCs in six plum cultivars (harvested from Seville, Spain). In addition, fruit varieties can strongly affect the profile of volatile components; therefore, investigating the differences in volatiles of different fruit varieties is extremely valuable for identifying different origins (genetic or geographic). Wu et al. [21] reported a difference in the volatile profiles of peaches using the HS-SPME-GC-MS method; a total of 37, 46, 41, 31, 38, and 34 volatiles were present in the fruits of separate peach cultivars, and the compositions and concentrations of VOCs endowed each peach cultivar with distinctive flavors. Apparent differences were also observed in the VOC profiles of the different plums; a total of 241, 254, 258, 236, and 243 volatiles were present in the ‘AL’, ‘XT’, ‘XL’, ‘ZH’, and ‘SX’ cultivars, respectively. To evaluate the metabolomic differences between groups and the variation status among the three replicates, we performed principal component analysis (PCA) of all samples. The PCA plot showed that all three replicates clustered together (Figure 5c). The traditional cultivars ‘XL’, ‘XT’, and ‘AL’ were clearly segregated into three separate groups, indicating large differences in volatile metabolites between groups. The new cultivars ‘SX’ and ‘ZH’ were not distinctly separated in the PCA, suggesting low variability in their volatile metabolites.

#### 3.6.2. Identification and Analysis of Differential Metabolites

To better define the most representative volatile compounds among the different plum cultivars, OPLS-DA, a multivariate statistical analysis, was employed to identify potential volatile markers for which the criteria were set to satisfy the conditions of VIP > 1 and *p* < 0.05 [53]. According to this, a total of 40 key volatile metabolites were screened, belonging to seven categories: 27.5% esters, 22.5% alcohols, 22.5% aldehydes, 15% terpenes, 7.50% ketones, and 5.00% acids (Figure 6a). A heat map hierarchical clustering analysis was performed on the 40 differential marker compounds to better characterize the flavor profiles of the five groups (Figure 6b). Each colored cell on the map corresponds to a relative concentration value, with volatile compounds in the rows and different cultivars in the columns. The rows were reordered according to the hierarchical clustering result, placing similar observations close to each other. In general, the volatile compounds in the five cultivars exhibited different profiles. The differential compounds were divided into two main categories, with each subdivided into two subcategories. The four subcategories were enriched in different plum varieties and labeled as characteristic volatile flavor substances. In detail, twelve volatile compounds (seven esters, two alcohols, one terpenoid, and two aldehydes) were identified as candidate markers in the traditional cultivar ‘AL’. Most ethyl esters (ethyl acetate, ethyl propionate, ethyl butanoate, and ethyl 2-methylbutanoate), along with acetate esters (n-propyl acetate and isobutyl acetate), were correlated with the cultivar ‘AL’, and these esters were also identified in a ‘Mobola plum’ from Southern Africa [54]. Ethyl esters are reported to be particularly important because they have a highly positive effect on consumer preference, have low odor thresholds, and provide fruity and floral notes [55]. Ethyl 2-methylpentanoate has been found in durians and is associated with specific flavors [54]. Aldehydes and alcohols, which are the dominant aroma components in plums, provide an overall green odor. In our results, six aldehydes (2-hexenal, 2-octenal, 2,4-heptadienal, nonanal, pentenal, and (Z)-2-decenal), four terpenoids (p-mentha-1,3,8-triene, (Z)-linalool oxide, hotrienol, and α-ionone), and three ketones (1-octen-3-one, 3-pentanone, and pentanone) were positively correlated with the traditional cultivar ‘XT’. Among them, 2-hexenal, heptanal, octanal, and nonanal have been described previously for different *Prunus salicina* Lindl. and contribute to the overall green odor and fruity aroma of plums [52,56,57]. In addition, α-ionone, (Z)-linalool oxide, and hotrienol contribute to the woody, violet-like, fruity odor of plums. Similarly, the traditional cultivar ‘XL’ had prominent green, apple-like, orchid, and fruity flavors because it contained four alcohols (2-phenylethanol, 1-octanol, 2-ethyl-1-hexanol, and nonanol) and two terpenoids (hotrienol and linalool) as candidate markers. In the new cultivar ‘ZH’, linalool, nonanol, (Z)-3-hexenol, nonanal, and hexanal were identified as candidate markers. Hexanal and (Z)-3-hexenol were key VOCs in the creation of the distinctive plum-like scent and were also found in the cultivar ‘SX’, providing the overall green odor [52]. In addition, hexyl acetate and hexenyl acetate were also considered markers of ‘SX’ and provided a fruity and sweet odor. Overall, esters, aldehydes, and alcohols were the key volatiles that differentiated the five plums.

## 4. Conclusions

A comparative analysis of the research data on the physicochemical indices, functional characteristics, and volatile flavor components of different plum cultivars from Sichuan was carried out to reveal their potential commercial value. These results suggest that all the investigated plums were characterized by a good taste, mainly due to their low acid–high sugar characteristics. The total sugar content of the five plums varied between 82.19 mg/g FW and 141.38 mg/g FW, and the total organic acid content ranged from 13.32 mg/g FW to 30.35 mg/g FW. The sweetness was mainly determined by sucrose (29.22–52.04%), and malic acid (33.21–46.78%) and quinic acid (40.59–52.44%) were the dominant contributors to the acidity of the five plums. All of the plums exhibited high levels of phenolic compounds, total flavonoids, and strong antioxidant properties. The radical scavenging capacity of the five plums measured using ABTS and DPPH assays ranged from 17.17 to 54.34 µmol TEs/g and 27.16 to 21.71 µmol TEs/g, respectively, and the FRAP reducing capacity of the plums varied between 7.75 and 32.19 µmol TEs/g. Among the five plum cultivars, the new cultivar ‘ZH’ had the highest TPC (271.74 mg of GAE/100 g) and TFC (169.84 mg of GAE/100 g) as well as strong antioxidant capacity. Furthermore, the characteristics of the volatile compounds in the different plum cultivars were investigated. The VOCs of the plums consisted of 31 acids, 106 alcohols, 48 aldehydes, 75 alkanes, 93 alkenes, 111 aromatic compounds, 76 ketones, 108 esters, 9 furans, and 80 other compounds. Among them, alcohols and esters were the major components in the plums. A total of 44 potential volatile markers related to the five plum cultivars were identified using our model. The major sweet fruit aroma contributors included ethyl acetate, isobutyl acetate, ethyl propionate, ethyl 3-methylbutanoate, and isoamyl acetate, which were identified in ‘AL’, and hexenyl acetate, butyl acetate, and hexyl acetate in ‘SX’. Aldehydes and alcohols were associated with green odors, such as 2-hexenal, 2-octenal, 2,4-heptadienal, nonanal, pentenal, and (Z)-2-decenal in ‘XT’; nonanol, hexanal, and (Z)-3-hexenol in ‘ZH’; and 2-phenylethanol, 1-octanol, 2-ethyl-1-hexanol, and nonanol in ‘XL’. This study demonstrates the quality characteristics of the important commercial cultivars of plum fruit in Sichuan and contributes to their wider use in breeding practice. Among the five plums, the new cultivar ‘ZH’ could be used as a source of new properties when improving the quality properties of plums due to its outstanding antioxidant activity and good taste. Furthermore, this study is an initial analysis of the volatile profile of plums from Sichuan; in future research, we will clarify the sensory contribution of volatiles to the overall aroma of plums and identify new markers based on different varietal genotypes.

## Figures and Tables

**Figure 1 foods-12-03801-f001:**
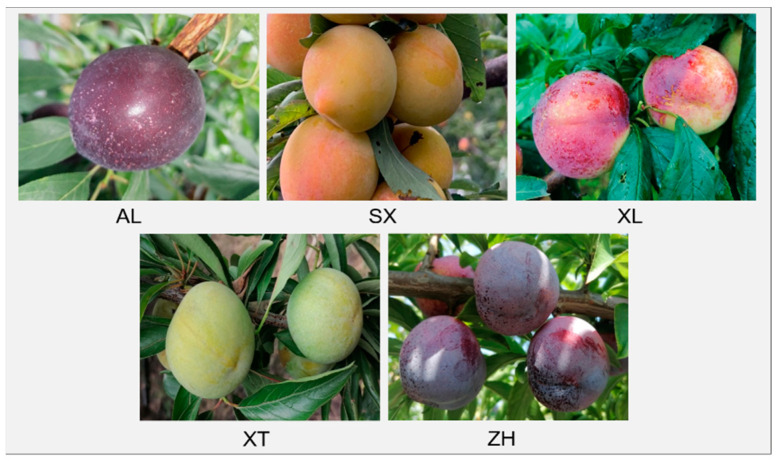
Sample photos of five plum cultivars. AL—ailisi; SX—shengxuepo; XL—xiangli; XT—xiangtian; ZH—zihuang.

**Figure 2 foods-12-03801-f002:**
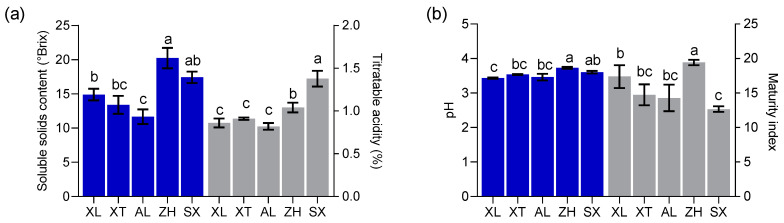
Soluble solids content (SSC) and titratable acidity (TA) (**a**) and pH and maturity index (MI) (**b**). All values are expressed as means ± SEM (*n* = 3). The mean values in each column with dissimilar letters are significantly different among the cultivars (*p* < 0.05). AL—ailisi; SX—shengxuepo; XL—xiangli; XT—xiangtian; ZH—zihuang.

**Figure 3 foods-12-03801-f003:**
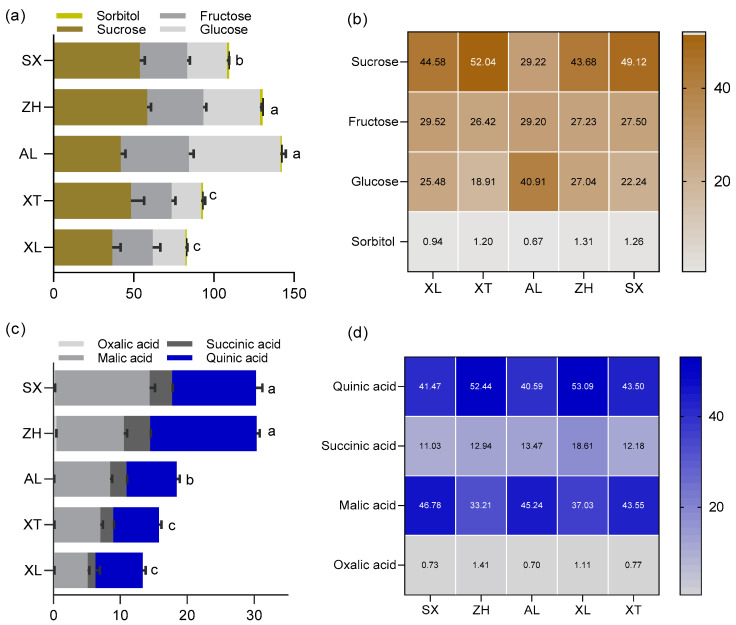
The content and composition of sugars (**a**,**b**), and the content and composition of organic acids (**c**,**d**) among the five plum cultivars. All values are expressed as means ±SEM (*n* = 3). The mean values in each column with dissimilar letters are significantly different among the cultivars (*p* < 0.05). AL—ailisi; SX—shengxuepo; XL—xiangli; XT—xiangtian; ZH—zihuang.

**Figure 4 foods-12-03801-f004:**
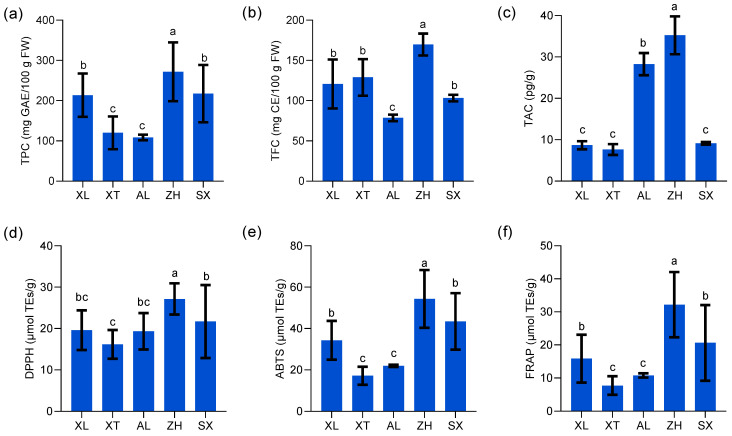
Total phenolic content (TPC) (**a**), total flavonoid content (TFC) (**b**), total anthocyanin content (TAC) (**c**), 2,2-diphenyl-1-picrylhydrazyl (DPPH) (**d**), 2,2′-azinobis (3-ethylbenzothiazoline-6-sulfonic acid) diammonium salt (ABTS) (**e**), and ferric reducing antioxidant power (FRAP) (**f**) of five plums cultivars. All values are expressed as means ±SEM (*n* = 3). The mean values in each column with dissimilar letters are significantly different among the cultivars (*p* < 0.05). AL—ailisi; SX—shengxuepo; XL—xiangli; XT—xiangtian; ZH—zihuang.

**Figure 5 foods-12-03801-f005:**
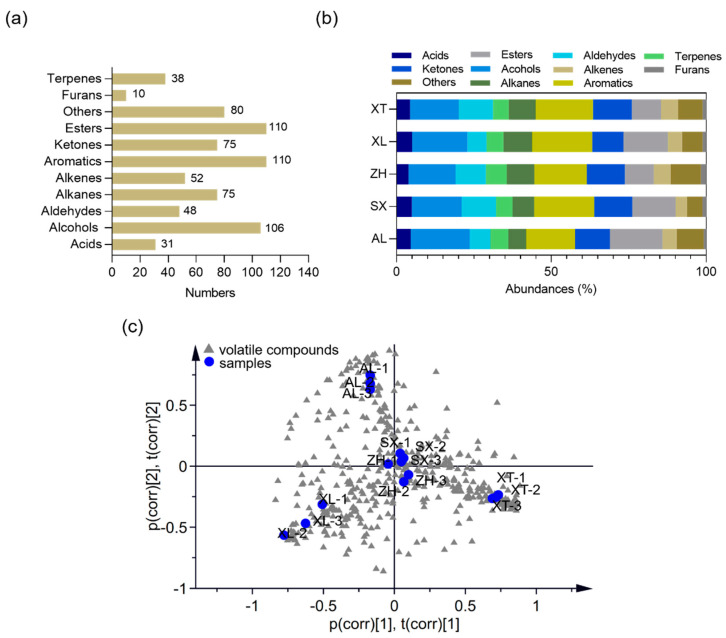
Volatile compounds of the five plum cultivars. Number (**a**) and composition (**b**) of volatile metabolites. (**c**) Principal component analysis (PCA) score plot for volatile metabolites (R^2^X = 0.814, Q^2^ = 0.767). AL—ailisi; SX—shengxuepo; XL—xiangli; XT—xiangtian; ZH—zihuang.

**Figure 6 foods-12-03801-f006:**
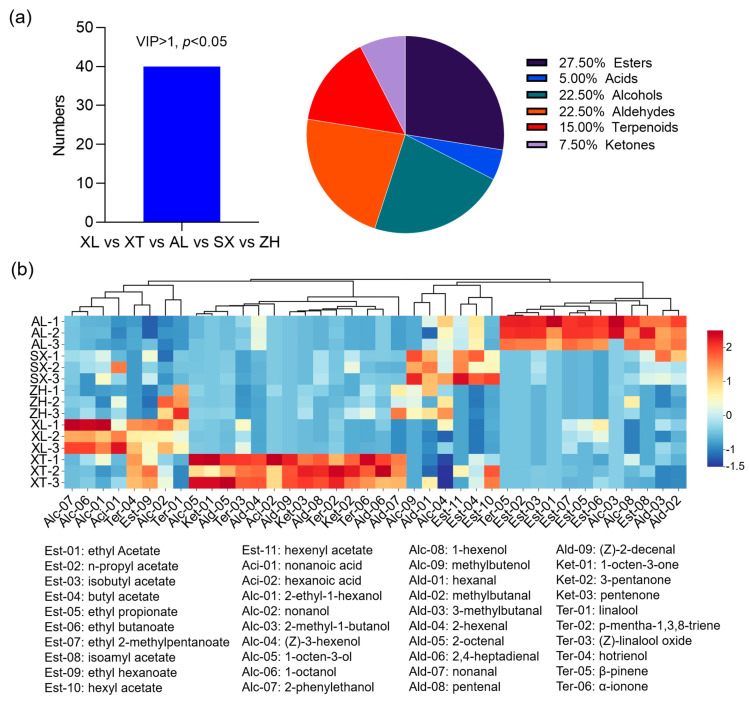
Characteristic volatile compound profiles in the five plum cultivars. Number and percentage of volatile compound subclasses (**a**); heat map of the selected major volatile compounds (**b**) in the five plum cultivars. Red and blue indicate up and down, respectively, and the darker the color, the higher the regulation level. VIP, variable importance in projection. AL—ailisi; SX—shengxuepo; XL—xiangli; XT—xiangtian; ZH—zihuang.

## Data Availability

Data are contained within the article.

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
