# Peer review of "The Physicochemical Attributes, Volatile Compounds, and Antioxidant Activities of Five Plum Cultivars in Sichuan"

_foods, 2023, doi:10.3390/foods12203801_

Round 1
Reviewer 1 Report
Report on the paper foods-2648579, see attached.
Report on the paper foods-2648579,
Physicochemical Attributes, Volatile Compounds, and Antioxidant Activities of Five Plum Cultivars in Sichuan
The manuscript has been well written (coherence, flow, and readability).
The main objective of this work was to study the physicochemical indices, functional characteristics and volatile flavor components of different Sichuan plum cultivars. This study demonstrates the quality characteristics of the important commercial cultivars of plum fruit in Sichuan and contributes to their wider use in breeding practice.
The title and abstract are appropriate for the content of the text.
The authors have an extensive research, with many results, regarding physicochemical characteristics, antioxidant capacity, and volatile profiles of five traditional and new plum cultivars in Sichuan, I enjoyed reading it.
On basis of these arguments, I can recommend this paper for publication in foods.
Author Response
List of Responses of Foods 2648579
Dear Reviewer 1:
Thank you for comments on MS entitled “Physicochemical Attributes, Volatile Compounds, and Antioxidant Activities of Five Plum Cultivars in Sichuan” (ID: Foods 2648579). The comments are very helpful for improving the MS.
Referee: 1
The manuscript has been well written (coherence, flow, and readability).The main objective of this work was to study the physicochemical indices, functional characteristics and volatile flavor components of different Sichuan plum cultivars. This study demonstrates the quality characteristics of the important commercial cultivars of plum fruit in Sichuan and contributes to their wider use in breeding practice. The title and abstract are appropriate for the content of the text. The authors have an extensive research, with many results, regarding physicochemical characteristics, antioxidant capacity, and volatile profiles of five traditional and new plum cultivars in Sichuan. I enjoyed reading it. On basis of these arguments, I can recommend this paper for publication in foods.
The authors’ answer: Thank you very much for your professional review. All of your suggestions and recognition are important guidance for our thesis writing and research work!
Once again, we sincerely thank the reviewer for their dedicated work.
Yours sincerely,
Zixi Lin
E-mail address: linzx1011@126.com

Reviewer 2 Report
The manuscript entitled “Physicochemical Attributes, Volatile Compounds, and Antioxi-2 dant Activities of Five Plum Cultivars in Sichuan” describes the physicochemical characteristics, antioxidant capacity, and volatile profiles of five plum varieties from China.
The manuscript is well organized, the objectives are clear, the results are well presented and the conclusions are supported by the results.
Author Response
List of Responses of Foods 2648579
Dear Reviewer 2:
Thank you for comments on MS entitled “Physicochemical Attributes, Volatile Compounds, and Antioxidant Activities of Five Plum Cultivars in Sichuan” (ID: Foods 2648579). The comments are very helpful for improving the MS.
Referee: 2
The manuscript entitled “Physicochemical Attributes, Volatile Compounds, and Antioxidant Activities of Five Plum Cultivars in Sichuan” describes the physicochemical characteristics, antioxidant capacity, and volatile profiles of five plum varieties from China. The manuscript is well organized, the objectives are clear, the results are well presented and the conclusions are supported by the results.
The authors’ answer: We appreciate your effort to review our manuscript, and your positive feedback. All of your suggestions and recognition are important guidance for our thesis writing and research work!
Once again, we sincerely thank the reviewer for their dedicated work.
Yours sincerely,
Zixi Lin
E-mail address: linzx1011@126.com

Reviewer 3 Report
The Introduction provides an overview of the importance of plums, particularly in Sichuan, China, and outlines the characteristics and qualities of various plum cultivars. It is generally clear and well-structured. It introduces the topic, discusses the significance of plums in different regions, and highlights the diversity of plum cultivars. However, there are some abrupt transitions between topics that could be smoother.
In summary, the paper effectively presents and analyses data on the physicochemical properties of different plum cultivars. It compares the values, discusses their significance for consumer acceptance, and references relevant research, making it a well-rounded and informative segment of a research report. It presents data and analysis related to sugar content and composition in plum cultivars. It compares the cultivars, discusses the implications for sweetness, and references relevant research, making it a valuable segment of a research report.
It provides context through references to previous studies, explains sources of variability, acknowledges methodological challenges, and highlights the importance of the phenolic compounds in fruit coloration and nutritional value.
The paper effectively presents data on volatile compounds in plum cultivars, describes the analytical approach, and references relevant studies. However, it could benefit from a more detailed interpretation of the results and a clearer connection to the study's objectives.
In my opinion, the article is very well-structured and comprehensive, requiring only minor revisions.
The English is quite good both grammatically and structurally, requiring no modifications.
Author Response
List of Responses of Foods 2648579
Dear Reviewer 3:
Thank you for comments on MS entitled “Physicochemical Attributes, Volatile Compounds, and Antioxidant Activities of Five Plum Cultivars in Sichuan” (ID: Foods 2648579). The comments are very helpful for improving the MS. We have studied comments carefully and have made correction which we hope meet with approval. Revised portion are marked in blue in revised MS. The responds to your comments are as follows:
Referee: 3
The Introduction provides an overview of the importance of plums, particularly in Sichuan, China, and outlines the characteristics and qualities of various plum cultivars. It is generally clear and well-structured. It introduces the topic, discusses the significance of plums in different regions, and highlights the diversity of plum cultivars. However, there are some abrupt transitions between topics that could be smoother. In summary, the paper effectively presents and analyses data on the physicochemical properties of different plum cultivars. It compares the values, discusses their significance for consumer acceptance, and references relevant research, making it a well-rounded and informative segment of a research report. It presents data and analysis related to sugar content and composition in plum cultivars. It compares the cultivars, discusses the implications for sweetness, and references relevant research, making it a valuable segment of a research report. It provides context through references to previous studies, explains sources of variability, acknowledges methodological challenges, and highlights the importance of the phenolic compounds in fruit coloration and nutritional value. The paper effectively presents data on volatile compounds in plum cultivars, describes the analytical approach, and references relevant studies. However, it could benefit from a more detailed interpretation of the results and a clearer connection to the study's objectives. The English is quite good both grammatically and structurally, requiring no modifications. In my opinion, the article is very well-structured and comprehensive, requiring only minor revisions.
The authors’ answer: Thanks for the full recognition and revision advice of MS. “Fruit flavour and nutrition are important factors in fruit quality” was added in line 43 to make the topic better connected. In addition, volatiles of five plum cultivars were determined and screening volatile markers for cultivar distinction in this study. All metabolites identified, as well as the distribution of subclasses across cultivars; further characterisation of differences among cultivars, including the combination of algorithms to screen for differential metabolites and their distribution across cultivars has been detailed in results (section 3.6.1 and 3.6.2). However, there was a lack of interpretation connection to the study's objectives, and the revision was marked bule in line 440 - 441.
Except revisions mentioned by reviewer, the other revisions were also marked blue in revised MS.
We tried our best to improve the manuscript. These improving will not influence the content and framework of the paper. We appreciate for Reviewer’s warm work earnestly, and hope that the correction will meet with approval.
Once again, thank you very much for your comments and suggestions.
Yours sincerely,
Zixi Lin
E-mail address: linzx1011@126.com

Reviewer 4 Report
in: Material and Methods
- the moisture content (%) of the fruits, the method for its determination and the conditions of the drying should be specified.
- the size of the ground fruits should be specified.
- it should be specified who makes the modifications of the methods by Wu et al. [21] and by Özdemir et al. [25].
- lines 113, 142 and 159: it should be specified what “room temperature” is.
in: Results and Discussions
- (30.35 and 30.26 g/kg) should be written instead of (30.35 and 30.26 g kg−1).
- line the word “etc.” should be written in italic.
- Drogoudi and Pantelidis [29] should be written instead of Drogoudi et al. [29]
- Tian and Schaich [52] should be written instead of Tian et al. [52].
- below the figures should be described the abbreviations: AL – ailisi; SX – shengxuepo; XL – xiangli; XT – xiangtian; ZH – zihuang.
- (Z)-linalool oxide should be written instead of (z)-linalool oxide; (Z)-2-decenal should be written instead of (z)-2-decenal, etc.
in: References
- No. 4: Walkowiak-Tomczak should be written instead of Walkowiak-tomczak.
- No. 18: the words “in vivo” and “in vitro” should be written in italic.
Author Response
List of Responses of Foods 2648579
Dear Reviewer 4:
Thank you for comments on MS entitled “Physicochemical Attributes, Volatile Compounds, and Antioxidant Activities of Five Plum Cultivars in Sichuan” (ID: Foods 2648579). The comments are very helpful for improving the MS. We have studied comments carefully and have made correction which we hope meet with approval. Revised portion are marked in blue in revised MS. The responds to your comments are as follows:
Referee: 4
1. Material and Methods
(1) The moisture content (%) of the fruits, the method for its determination and the conditions of the drying should be specified.
The authors’ answer: Thank you very much for your comments. As the results determined in this paper were all for fresh fruit content without the use of drying techniques. To avoid any misinterpretation of your reading, we have revised the description in lines 89 - 90.
(2) The size of the ground fruits should be specified.
The authors’ answer: Thanks for the comments. Done accordingly and the related information was added in line111 and marked in blue.
(3) It should be specified who makes the modifications of the methods by Wu et al. [21] and by Özdemir et al. [25].
The authors’ answer: Thank you very much for your valuable comments. In order to avoid logical errors in statements, we have reorganized the language. The revised parts were marked in blue.
(4) Lines 113, 142 and 159: It should be specified what “room temperature” is.
The authors’ answer: Thanks for the comments. Done accordingly and the related information was added in line 114, 143 and 160, and marked in blue.
2. Results and Discussions
(1) (30.35 and 30.26 g/kg) should be written instead of (30.35 and 30.26 g kg−1).
The authors’ answer: Thanks for the comments. Done accordingly and marked in blue.
(2) Line the word “etc.” should be written in italic.
The authors’ answer: Thanks for the comments. Done accordingly and marked in blue.
(3) Drogoudi and Pantelidis [29] should be written instead of Drogoudi et al. [29].
The authors’ answer: Thanks for the comments. We have changed “Drogoudi et al. [29]” to “Drogoudi and Pantelidis [29]” in MS.
(4) Tian and Schaich [52] should be written instead of Tian et al. [52].
The authors’ answer: Thanks for the comments. Done accordingly and marked in blue.
(5) Below the figures should be described the abbreviations: AL – ailisi; SX – shengxuepo; XL – xiangli; XT – xiangtian; ZH – zihuang.
The authors’ answer: Thank you very much for your valuable suggestions. To address the reviewer’s comments, we have added the information below all figures in the revised MS.
(6) (Z)-linalool oxide should be written instead of (z)-linalool oxide; (Z)-2-decenal should be written instead of (z)-2-decenal, etc.
The authors’ answer: Thanks for the comments. Done accordingly.
3. References
(1)Walkowiak-Tomczak should be written instead of Walkowiak-tomczak.
The authors’ answer: Thanks for the comments. Done accordingly.
(2)NO. 18: the words “in vivo” and “in vitro” should be written in italic.
The authors’ answer: Thank you very much for your comments. Accordingly, we have changed the words “in vivo” in the MS to italic.
Except revisions mentioned by reviewer, the other revisions were also marked blue in revised MS.
We tried our best to improve the manuscript. These improving will not influence the content and framework of the paper. We appreciate for Reviewer’s warm work earnestly, and hope that the correction will meet with approval.
Once again, thank you very much for your comments and suggestions.
Yours sincerely,
Zixi Lin
E-mail address: linzx1011@126.com
